# OpenReview forum: "Adaptive Deployment of Untrusted LLMs Reduces Distributed Threats"
_ICLR.cc/2025/Conference — ICLR 2025 Poster_

### Official Review · Reviewer_wCeL · 2024-11-04

**Soundness:** 3
**Presentation:** 3
**Contribution:** 3
**Rating:** 8
**Confidence:** 3

**Summary:**

The paper proposes a framework for safely deploying untrusted large language models (LLMs) by introducing Diffuse Risk Management to balance usefulness and safety in scenarios where subtle misalignments may go undetected. The approach involves two layers: micro-protocols to manage task-level interactions and macro-protocols to dynamically adjust the selection of micro-protocols based on the model’s observed behavior. This setup is evaluated in a red team vs. blue team setting, particularly on code generation tasks where backdoor vulnerabilities are plausible risks. The proposed methods reportedly improve both safety and usefulness tradeoffs compared to baseline approaches by continuously updating the model’s trust assessment and balancing between a powerful but risky model and a weaker, safer one.

**Strengths:**

- The research question is interesting and important, which is a valuable addition for safe LLM deployment.

- The red team vs. blue team evaluation on code generation tasks effectively demonstrates the framework's performance and practical relevance.

- Results show significant gains over baselines, indicating that the adaptive framework achieves a good balance between utility and safety.

**Weaknesses:**

- The layered approach of micro-protocols for individual tasks and a macro-protocol for overall risk adjustment is promising, but the macro-protocol’s Bayesian update mechanism needs further clarification. Detailing how credence values are updated based on safety scores would make this process easier to replicate. Including pseudo-code or equations for these updates, specifically on initial priors and how evidence from each task influences credences, would enhance clarity. Additionally, the concept of a “mixed-strategy Nash equilibrium” to model interactions between red and blue teams is mentioned briefly; a more thorough integration of this equilibrium into the protocol's design would improve its practical relevance.

- Relying on a pre-defined reference distribution of aligned and misaligned outputs may limit scalability, particularly for rapidly evolving or open-ended tasks. Information on how this distribution was generated (e.g., sample size, diversity, labeling criteria) would enhance transparency, especially if future tasks diverge from those tested here. Discussing alternative strategies, like dynamically updating the reference set or using sampling techniques, could make this approach more feasible for real-world applications.

- The safety-usefulness tradeoff is well-explored, yet the effects of varying the safety coefficient (α) on model behavior would be worth examining. Showing how different α values impact the macro-protocol’s balance between safety and utility would clarify its adaptability to different risk levels. Also, adding performance comparisons across different red team tactics or over varying task sequence lengths could highlight whether the adaptive protocol consistently outperforms static approaches or if its advantages diminish in certain scenarios.

- The red team’s static attack strategy does not capture adaptive, iterative evasion that may be expected from real-world adversaries. Enabling the red team to adjust its strategy based on blue team responses would create a more rigorous test of the framework’s defenses. Additionally, although “sneakiness” is used to rate adversarial examples, it remains unclear how this is quantified or validated. Defining sneakiness in more detail, perhaps with examples of high- vs. low-sneakiness backdoors, would improve the credibility of the threat model.

- The reliance on a significant capability gap between trusted and untrusted models raises concerns about scalability. Testing the framework with a smaller gap, such as using different versions of similar models or narrowing the gap incrementally, would shed light on its robustness. Showing how detection accuracy varies with smaller capability gaps would help determine if this approach holds up as trusted models begin to approximate the untrusted model’s performance, a likely future scenario.

- While the approach is well-suited to code generation, other domains like recommendation systems or content moderation involve more subjective forms of misalignment. Discussing how the micro- and macro-protocols could adapt to detect implicit or subtle safety risks in these fields would broaden the framework’s applicability. Insights into handling subjective safety evaluations, where misalignment is not as binary as in code, would show how this approach could extend to tasks with different alignment requirements.

- A minor issue in the paper is a broken reference in Appendix G section, specifically in Finding 2, where the reference to the intended figure is missing (“Figure ??”).

**Questions:**

- How does the macro-protocol handle varying types of misalignment or malicious behavior in adaptive red team settings?

---

> ### Author Response · Authors · 2024-11-18
> **Response to Reviewer wCeL (1/2)**
>
> We thank the reviewer for their remarks and are glad they noted the importance of the setting and the effectiveness of the evaluation. We respond in line to your responses and are happy to expand or provide further responses if necessary.
>
> > the macro-protocol’s Bayesian update mechanism needs further clarification. Detailing how credence values are updated based on safety scores would make this process easier to replicate. Including pseudo-code or equations for these updates, specifically on initial priors and how evidence from each task influences credences, would enhance clarity.
>
> We have revised section 3.2 to make these details clear.
>
> > Additionally, the concept of a “mixed-strategy Nash equilibrium” to model interactions between red and blue teams is mentioned briefly; a more thorough integration of this equilibrium into the protocol's design would improve its practical relevance.”
>
> We have included additional details in the updated section 3.2 and added two new appendices to explore these elements in more detail. Appendix A explains how finding the Nash equilibrium chronologically fits into the process of a control evaluation, and Appendix B.2 explains how this Nash equilibrium is used to construct the initial prior of the macro-protocol.
>
> > “Relying on a pre-defined reference distribution of aligned and misaligned outputs may limit scalability, particularly for rapidly evolving or open-ended tasks. Information on how this distribution was generated (e.g., sample size, diversity, labeling criteria) would enhance transparency, especially if future tasks diverge from those tested here.
>
> We agree with the reviewer that access to a reference distribution of aligned and misaligned tasks is a key affordance for the blue team. However, we argue in (newly added) Appendix B.3 that this affordance is reasonable, as constructing such a reference distribution is possible by prompting or fine-tuning future untrusted models and evaluating their harmfulness via time-intensive manual evaluations.
>
> To address the lack of detail about our reference distribution, we have expanded our discussion in section 3.2 (macro-protocol) and added Appendix B.3, which clarifies the data used in the reference distribution and explores how model deployers could generate such reference distributions for future deployments.
>
> > Discussing alternative strategies, like dynamically updating the reference set or using sampling techniques, could make this approach more feasible for real-world applications.
>
> We think this is a great idea -- dynamically updating the reference set with recent tasks from deployment seems very promising for deployments which change over time.  While our chosen setting doesn’t require this, we have added it to the (new) discussion of reference distributions in Appendix B.3 as a way to extend our work to evolving settings (see the last paragraph of Appendix B.3).
>
> > The safety-usefulness tradeoff is well-explored, yet the effects of varying the safety coefficient (α) on model behavior would be worth examining. Showing how different α values impact the macro-protocol’s balance between safety and utility would clarify its adaptability to different risk levels.
>
> We apologize for any confusion - in Figure 3 and Figure 4 (middle), each point in one safety-usefulness pareto curve corresponds to a distinct value of the safety coefficient α.  We’ve adjusted the Figure 3 caption accordingly to clarify this.  To show in further detail how α affects the macroprotocol’s decisions, here is a table showing how often each micro-protocol was chosen at three different values of alpha.  The usage of Query Trusted increases as α increases, leading to higher safety but lower usefulness.
> | $\alpha$    | Usage of Query Untrusted | Usage of Query Trusted | Usage of Other Micro-protocols | **Safety** | **Usefulness** |
> | ---- | ------------------------ | ---------------------- | ------------------------------ | ---------- | -------------- |
> | 1    | 75%                      | 14%                    | 12%                            | 99.24%     | 46.7%          |
> | 20   | 36%                      | 56%                    | 9%                             | 99.88%     | 37.8%          |
> | 100  | 7%                       | 92%                    | 2%                             | 99.99%     | 27.1%          |
>
> > Also, adding performance comparisons across different red team tactics or over varying task sequence lengths could highlight whether the adaptive protocol consistently outperforms static approaches or if its advantages diminish in certain scenarios.
>
> Figure 3 (right) shows the performance of our adaptive macro-protocol at different task sequence lengths (from 10 through 10000 tasks). Our macro-protocol already dominates the static baselines over 10 tasks, and consistently improves given longer task sequences.

---

> ### Author Response · Authors · 2024-11-18
> **Response to Reviewer wCeL (2/2)**
>
> > Enabling the red team to adjust its strategy based on blue team responses would create a more rigorous test of the framework’s defenses.
>
> Please see [General Response 2](https://openreview.net/forum?id=keu6sxrPWn&noteId=gT8AsebBRW).
>
> > Defining sneakiness in more detail, perhaps with examples of high- vs. low-sneakiness backdoors, would improve the credibility of the threat model.
>
> We apologize for the lack of clarity around the definition of “sneakiness”. In our work, sneakiness refers to the untrusted model’s estimate of the monitor score for a given malicious output. Thus, a higher-sneakiness malicious output means the untrusted model thinks it will be more likely to bypass the monitor. Figure 4 (right) additionally explores how well the “sneakiness” estimate correlates to the actual monitor score provided by the trusted model. This figure has been updated from the original draft to be more consistent with the definitions of monitor score and sneakiness used in the paper.
> We have updated section 3.2 to clarify the definition of sneakiness. We presented examples of high and low sneakiness backdoors to Appendix F.
>
> > The reliance on a significant capability gap between trusted and untrusted models raises concerns about scalability. Testing the framework with a smaller gap, such as using different versions of similar models or narrowing the gap incrementally, would shed light on its robustness. Showing how detection accuracy varies with smaller capability gaps would help determine if this approach holds up as trusted models begin to approximate the untrusted model’s performance, a likely future scenario.
>
> First, we would like to clarify that model deployment becomes more challenging as the capability gap between trusted and untrusted models becomes **larger**  instead of **smaller**.
> To explore the capability gap, we have added new results to Figure 3. Please see [General Response 1](https://openreview.net/forum?id=keu6sxrPWn&noteId=gT8AsebBRW) for more discussion.
>
> > While the approach is well-suited to code generation, other domains like recommendation systems or content moderation involve more subjective forms of misalignment. Discussing how the micro- and macro-protocols could adapt to detect implicit or subtle safety risks in these fields would broaden the framework’s applicability. Insights into handling subjective safety evaluations, where misalignment is not as binary as in code, would show how this approach could extend to tasks with different alignment requirements.
>
> We want to clarify that our approach is not specifically tailored for code generation. In particular, we do not use domain-specific features like a code interpreter or test case execution to detect malicious outputs. Instead, our proposed combination of micro-protocols and the adaptive macro-protocol can adapt to other common LM application domains such as text generation, where misinformation is an example of a malicious output.
>
> For domains where safety evaluations are subjective, we agree that there are novel domain-specific challenges in defining a safety taxonomy. However, this is outside the scope of our paper, which focuses on establishing a general and domain-agnostic deployment framework of untrusted models.
>
> Finally, we thank the reviewer again for their comments and urge them to reconsider their score in light of our response and the other reviews.

---

> ### Author Response · Authors · 2024-11-24
> **Looking forward to your comment**
>
> We have added further clarifications and experiments to address your reviews. Please feel free to let us know if you have any questions. Thanks a lot!!

---

### Official Review · Reviewer_QoGP · 2024-11-04

**Soundness:** 3
**Presentation:** 3
**Contribution:** 4
**Rating:** 8
**Confidence:** 3

**Summary:**

The paper takes inspiration from Paul Christiano's definition of a "low stakes" scenario and defines the problem of Diffuse Risk Management -- where the goal is to deploy a strong, but untrusted LLM to maximize usefulness while minimizing harm. This setting is very interesting and the paper clearly defines the problem formally. The paper then proposes a solution to this problem by decomposing the solution into a pair made up of (a) macro protocol and (b) a set of micro protocols. The micro protocols act at a task/problem instance level and output the input's answer along with an estimate of the safety score (based on a weaker but trusted model). The macro protocol employs this set of micro protocols and chooses between these responses based on the track record of the powerful but untrusted model. The macro protocol learns to estimate the usefulness/safety of a response based on an evolving set of hypotheses that get updated using a bayesian inference framework. The paper shows strong empirical evidence of the effectiveness of their adaptive protocol over static baselines. The authors conclude by discussing the limitations and proposing interesting future direction of research.

**Strengths:**

1. The problem being introduced is very interesting and relevant. The paper defines the problem statement very clearly.
2. The decomposition into micro and macro protocols is compelling and well explored
3. The paper provides strong empirical analysis using several state of the art models.
4. The paper provides useful details the appendixes.
5. The writing is generally very clear* and engaging.

* barring some details in the equations and mathematical formulation

**Weaknesses:**

The mathematical formulation could be improved by providing some details and an example.
    - E.g., "To establish credence on U ’s alignment, our macro-protocol tracks hypotheses h_i" could be clarified to indicate that the number of hypotheses is not related to the number of micro protocols (unless I'm misunderstanding something).
    - Similarly, in Eq 7, it is unclear what s_i refers to as it isn't referred to anywhere else
    - The effect of number of hypotheses, and how the authors develop the reference distributions etc. are missing, and such details are important for future work to build up on this work.

The other main weakness I see, and the authors already acknowledge this as a future work item is exploring the capability gap. This is very important, especially in a low-stake scenario (as assumed by the paper) because as Paul Christiano points out ((https://ai-alignment.com/low-stakes-alignment-f3c36606937f)), it is important in this setting for the humans/monitoring systems to understand the outputs of the LLM, and if we use a weaker model, is it feasible to assume the weaker model can always detect a stronger model's subtly malicious intent?

**Questions:**

1. While somewhat intuitive, it'd be great if you could add a discussion around why it's reasonable to expect that a weak trusted model can exist in the first place -- i.e., if the model is trusted due to extensive testing, why can't the same testing be done for the larger model? (Answer: because the larger model can be more subtle about cheating the test -- but what's the guarantee that the weaker tested model isn't capable of doing the same?).

2. Nit: Eq (5) -- Why is x_t missing from the equation but is part of the description?

3. I don't fully understand how the hypotheses are defined, how many should one define and how does that quantity impact the performance of the system. Either an example or description of the heuristics would be very useful here.

4. What's the intuition behind adaptive protocol being _safer_ than static threshold protocol (Figure 2 [left])? E.g., i could set the static threshold to be very high such that it always triggers trusted model, which should ideally 'never' return unsafe code.


5. Does the red team have access to the model being queried? It's unclear because in the paper you say "To simulate the worst-case scenario, we grant the red team more affordance to adversarially attack the blue team and minimize safety: 1) given a micro-protocol used by the blue team..." -- does this imply access to the trusted model as well? If so, why would the red team have to estimate the blue team's monitor score? And if not, is it a reasonable threat model to consider?

---

> ### Author Response · Authors · 2024-11-18
> **Response to Reviewer QoGP**
>
> We thank the reviewer for their comments. We are happy they think the “problem being introduced is very interesting and relevant.” We address the two main comments you have and then answer specific clarification questions, and are happy to expand or provide further responses if necessary.
>
> > Mathematical formulation could be improved… I don't fully understand how the hypotheses are defined, how many should one define and how does that quantity impact the performance of the system.
>
> We thank the reviewer for their identification of our notational mistakes and the need to spell out more details. To address these, we have:
> - Rewritten section 3.2 and included an algorithm section in Appendix B.4 to make our method more explicit.
> - Added Appendix B.2 explaining in detail how the prior is constructed.  The last paragraph of B.2 explains where the number of hypotheses comes from.
>
> > The other main weakness I see, and the authors already acknowledge this as a future work item is exploring the capability gap.... if we use a weaker model, is it feasible to assume the weaker model can always detect a stronger model's subtly malicious intent?
>
> Please see [General Response 1](https://openreview.net/forum?id=keu6sxrPWn&noteId=gT8AsebBRW).
>
> > if the model is trusted due to extensive testing, why can't the same testing be done for the larger model?
>
> A model is considered trusted due to
> - Extensive real-world deployment and stress-testing (e.g., GPT-3.5 with a track record of safe operation). In contrast, the new, untrusted model lacks such testing before deployment. Our proposed method can help deployers gradually establish trust in the untrusted model during deployment.
> - Limited capability to coherently pursue hidden goals and bypass safety checks. However, the strong, untrusted model might have such capabilities.
>
> > Does the red team have access to the model being queried? It's unclear because in the paper you say "To simulate the worst-case scenario, we grant the red team more affordance to adversarially attack the blue team and minimize safety: 1) given a micro-protocol used by the blue team..." -- does this imply access to the trusted model as well? If so, why would the red team have to estimate the blue team's monitor score? And if not, is it a reasonable threat model to consider?
>
> This affordance does not allow the red team direct access to the trusted model (e.g., getting the exact blue team’s monitor score of its own outputs). We believe this is a more practical setting; otherwise, it requires the untrusted model to 1) identify the weak model, and the deployer’s monitor scaffolding, 2) replicate the same scaffolding and query the weak model itself.
>
> Please see [General Response 2](https://openreview.net/forum?id=keu6sxrPWn&noteId=gT8AsebBRW) for more discussions.
>
> > What's the intuition behind adaptive protocol being safer than static threshold protocol (Figure 3 [left])? E.g., i could set the static threshold to be very high such that it always triggers trusted model, which should ideally 'never' return unsafe code.
>
> The reviewer is correct that in the limit of high threshold / high safety coefficient, both protocols exclusively use the trusted model, achieving 100% safety and 24% usefulness.  What Figure 3 shows is that **at any given level of usefulness** above that, the adaptive protocol is safer because it can tailor its decisions to the level of misalignment.  The figure cannot extend to 100% safety due to the log scale of the x-axis, but both series could be extended further to the bottom-right with more datapoints.
>
> We thank the reviewer again for their comments and positive reviews and encourage them to continue supporting this paper.

---

> > ### Author Response · Authors · 2024-11-24
> > **Looking forward to your comment**
> >
> > We have added further clarifications and experiments to address your reviews. Please feel free to let us know if you have any questions. Thanks a lot!!

---

### Official Review · Reviewer_CDuS · 2024-11-04

**Soundness:** 3
**Presentation:** 3
**Contribution:** 3
**Rating:** 5
**Confidence:** 3

**Summary:**

The paper proposes a defense for safely using untrusted large language models (LLMs) that may generate harmful outputs. This defense involves a two-step process. First, a trusted, weaker model monitors the stronger, untrusted model’s outputs, checking each task for potential harm. In the second step, the system monitors the long-term performance of the untrusted model, dynamically adjusting the acceptance threshold based on the model’s reliability. If the model consistently performs safely, the threshold is raised, making rejections less likely. Experiments in a code-generation test environment show that this approach achieves high safety with minimal compromise to the untrusted model’s effectiveness.

**Strengths:**

- The topic of safely using untrusted models is interesting.
- The paper is well-structured.
- The idea of dynamically assigning thresholds is nice.

**Weaknesses:**

Some weaknesses in my view:

- The authors should consider the potential synergy between tasks, as this might allow more stealthy attacks.
- There should be discussion on how to defend against adaptive attacks, where an adversary gradually increases the harmfulness of tasks.
- The assumptions underlying the strength of trusted models need further clarification, including whether these assumptions are reasonable.

I have three main questions regarding the synergy between tasks, potential countermeasures against attackers, and the assumptions underlying the attack model, as outlined below:

The proposed method does not seem to account for the synergy between tasks. Specifically, an adversary might bypass defenses by performing harmful actions through a sequence of well-designed tasks. For instance, if transferring money from A to B is harmful, the model could break this action into two steps: first transferring money from A to C, and then from C to B. Overlooking task synergy could fail to detect such harmful intentions.

The core of the proposed defense is dynamically adjusting the threshold that determines whether a task is untrusted based on the model’s credibility, which is calculated from previous performance. In this case, an adversary could adapt their behavior, first establishing credibility by performing safe tasks, then gradually increasing the harmfulness of their actions while keeping them below the threshold. Eventually, the threshold might become high enough that the system perceives all actions from the adversary as harmless, despite potentially harmful intentions.

The authors assume that, alongside the untrusted strong model, the defender has access to a series of weak trusted models to assess harmfulness. However, this assumption requires further discussion. If the trusted models are too weak, the untrusted model could deceive them. Conversely, if they are too strong, it raises the question of why these trusted models are not simply used to perform the tasks directly. The experimental setting raises this concern further, as the authors use Claude-3-Opus as a "weak" trusted model, which performs almost as well as the untrusted “strong” model.

**Questions:**

see the section on Weaknesses above.

---

> ### Author Response · Authors · 2024-11-12
> **Weakness Section refers to different paper.**
>
> We want to flag both to the reviewer and the AC that we think the weaknesses section might be referring to another paper.
>
> For example:
>
> - We do not have a "theorem on the vulnerability of endogenous verification" in Section 3
> - We do not refer to "eve" at any point in our paper
> - We do not explain or explore the limits of self-verification in CTM
>
> We'd really appreciate the correct review as quickly as possible!

---

> > ### Comment · Reviewer_CDuS · 2024-11-12
> > **Correct part of the review uploaded**
> >
> > My sincere apologies for the copy&paste mistake. It is corrected now.

---

> ### Author Response · Authors · 2024-11-18
> **Response to Reviewer CDuS**
>
> We thank the reviewer for their remarks and are glad they noted the paper is well structured and that the setting is interesting. We respond in line to your responses and are happy to expand or provide further responses if necessary.
>
> > The authors assume that, alongside the untrusted strong model, the defender has access to a series of weak trusted models to assess harmfulness…... The experimental setting raises this concern further, as the authors use Clause-3-Opus as a "weak" trusted model, which performs almost as well as the untrusted “strong” model
>
> We would like to clarify that our main safety-usefulness results in Figure 3 are all based on using GPT-3.5-turbo as the weak trusted model, which we have also noted in the updated caption. Please see [General Response 1](https://openreview.net/forum?id=keu6sxrPWn&noteId=gT8AsebBRW) for more discussions about whether weaker trusted models can monitor untrusted models. Note that all of our trusted models including Claude-3-Opus have much lower performance than the untrusted model on our APPS subset (3% to 24% for trusted models vs. 47% for GPT-4).
>
> > The proposed method does not seem to account for the synergy between tasks… might bypass defenses by performing harmful actions through a sequence of well-designed tasks.
>
> What the reviewer suggests is a grouping of tasks such that a sequence of $N$ correlated actions can result in a significant failure (e.g. multiple correlated PRs).
>
> If each individual action is not entirely benign, our method remains effective. For example, if it requires $N=10$ sequential actions to cause a substantial failure, our method would become increasingly cautious about the untrusted model’s outputs, and thus might directly opt for the trusted model’s output.
>
> However, if each individual action is indeed entirely benign, we agree this is outside of the scope of the current paper. We will record this in the limitation section.
>
> >  In this case, an adversary could adapt their behavior, first establishing credibility by performing safe tasks, then gradually increasing the harmfulness of their actions while keeping them below the threshold
>
> Please see [General Response 2](https://openreview.net/forum?id=keu6sxrPWn&noteId=gT8AsebBRW).
>
>
> We thank the reviewer again for their comments and urge them to reconsider their score in light of our response and our updated manuscript.

---

> ### Author Response · Authors · 2024-11-24
> **Looking forward to your comment**
>
> We have added further clarifications and experiments to address your reviews. Please feel free to let us know if you have any questions. Thanks a lot!!

---

> > ### Author Response · Authors · 2024-12-03
> > **Request for response**
> >
> > Hey Reviewer CDuS,
> >
> > We'd really appreciate if you could read our specific response, the general response and the other reviewers remarks.
> > In particular, we'd really appreciate if you could double check this is your final score (as the review period ends today).
> >
> > Thanks in advance,
> > Authors

---

### Author Response · Authors · 2024-11-18
**General Response**

Dear Reviewers,

Thank you all for your thoughtful reviews! We are excited that the reviewers recognized the strength of our paper:

- **Interesting and important topic (all three reviewers):** “The topic of safely using untrusted models is interesting.”, “The problem being introduced is very interesting and relevant.”, “The research question is interesting and important, which is a valuable addition for safe LLM deployment.”
- **Novel and compelling method (CDuS, QoGP):** “The idea of dynamically assigning thresholds is nice.”, “the decomposition into micro and macro protocols is compelling and well explored”
- **Strong empirical results (QoGP, wCeL):** “The paper provides strong empirical analysis.”, “The red team vs. blue team evaluation effectively demonstrates the framework's performance and practical relevance.”
- **Well-structured, engaging writing (CDuS, QoGP):** “The paper is well-structured.”, “The writing is generally very clear* and engaging.”, “ provides useful details in the appendixes.”

We’ve addressed all comments individually and incorporated them into the main paper (now highlighted in blue text).

Here we address the shared concerns from reviewers.

### Can weaker trusted models monitor untrusted models?

Our empirical results indicate that safety monitoring is less challenging than problem-solving. Specifically, while there is a significant gap between Claude-3-Sonnet/GPT-3.5-turbo and GPT-4 in terms of the pass@1 in APPS (see table below), our results in Figure 4 (right) indicate that these weak models still perform well in monitoring. For example, GPT-3.5-turbo achieves an AuROC of 0.91 in monitoring GPT-4-generated backdoored code, and achieves >99.0% safety based on our method.

| Metric | Claude-3-Haiku | Claude-3-Sonnet | GPT-3.5-turbo | Claude-3-Opus | GPT-4 |
|--------|----------------|-----------------|---------------|---------------|--------------------|
| pass@1 | 3.1          | 13.2            | 23.9          | 23.9          | 47.3   |
| AuROC | 0.67         | 0.84            | 0.91          | 0.94          | N/A   |

To understand the impact of large capability gaps between trusted and untrusted models, we added results to Figure 4 showing safety-usefulness pareto frontiers for all four trusted models (**[here is an anonymous image link for convenience](https://ibb.co/wMyXfwv)**).
When the capability gap is moderately large (e.g. Claude-3-Sonnet/GPT-3.5-turbo vs GPT-4), our method still can effectively maintain > 98% usefulness of the untrusted model while ensuring a safety score > 99%.
However, when the capability gap is extremely large (e.g. Claude-3-Haiku vs GPT-4), we calculate a monitoring AuROC score of merely 0.67. This then leads to a substantially reduced safety-usefulness tradeoff: maintaining 80% usefulness of the untrusted model while ensuring a safety score > 95%.

### What are reasonable affordances for the red team?
All three reviewers have asked related questions:
- Reviewer CDuS: can the red team first establish credibility by performing safe tasks, then gradually increasing the harmfulness of their actions while keeping them below the threshold?
- Reviewer wCeL: can the red team adjust its strategy based on blue team responses?
- Reviewer QoGP: can the red team have access to the model being queried?

We have updated Lines 178-186 to clarify two restrictions on the red team which represent plausible real-world choices for safety-conscious deployers:
1. The red team strategy is not permitted to carry state between tasks (e.g. does not have memory to store previous blue team queries). This assumption is also made by Greenblatt et al. [1], who state that it is valid because “... the AI developer always gets to choose what context to provide to the LLM.” We concur -- since LLMs do not update their weights during inference, developers have control over whether they are stateful. In deployments where carrying state between tasks is useful but not strictly necessary, developers have the option of forbidding it for the sake of safety.
2. The red team does not have access to the trusted model (it cannot query the trusted model to test the suspiciousness of a malicious action it is considering). This is also realistic -- in a well-monitored deployment, the untrusted model cannot privately query the trusted model on malicious actions without those actions being seen by the monitor.

### More details about macro-protocols.

We refer the reviewers to our new section 3.2 where we have added additional details on the credence update and expected utility calculation. The new Appendix B explains the creation of reference distributions, derives an optimal prior, clarifies the role of Nash equilibrium, and gives pseudocode for our method.

[1] AI control: Improving safety despite intentional subversion. ICML2024

---

### Meta-Review · Area_Chair_Erkw · 2024-12-24

**Metareview:**

The paper proposes a defense for safely using untrusted large language models (LLMs) that may generate harmful outputs. The defense involves a two-step process. Experiments in a code-generation test environment show that this approach achieves high safety with minimal compromise to the untrusted model’s effectiveness. The authors are encouraged to include the discussion about adaptive attacks and the rationale of the trust small model in the final version.

**Additional Comments On Reviewer Discussion:**

The reviewers agree with the final decision.

---

### Decision · Program_Chairs · 2025-01-22

Accept (Poster)